# New Insight in Cardiorenal Syndrome: From Biomarkers to Therapy

**DOI:** 10.3390/ijms24065089

**Published:** 2023-03-07

**Authors:** Giovanna Gallo, Oreste Lanza, Carmine Savoia

**Affiliations:** Clinical and Molecular Medicine Department, Faculty of Medicine and Psychology, Sant’Andrea Hospital, Sapienza University of Rome, 00189 Rome, Italy

**Keywords:** heart failure, renal disease, SGLT2 inhibitors

## Abstract

Cardiorenal syndrome consists in the coexistence of acute or chronic dysfunction of heart and kidneys resulting in a cascade of feedback mechanisms and causing damage to both organs associated with high morbidity and mortality. In the last few years, different biomarkers have been investigated with the aim to achieve an early and accurate diagnosis of cardiorenal syndrome, to provide a prognostic role and to guide the development of targeted pharmacological and non-pharmacological therapies. In such a context, sodium-glucose cotransporter 2 (SGLT2) inhibitors, recommended as the first-line choice in the management of heart failure, might represent a promising strategy in the management of cardiorenal syndrome due to their efficacy in reducing both cardiac and renal outcomes. In this review, we will discuss the current knowledge on the pathophysiology of cardiorenal syndrome in adults, as well as the utility of biomarkers in cardiac and kidney dysfunction and potential insights into novel therapeutics.

## 1. Introduction

Cardiorenal syndrome (CRS) is characterized by the coexistence of acute or chronic dysfunction of heart and kidneys resulting in a cascade of feedback mechanisms causing damage to both organs [1]. Clinical and epidemiological studies highlighted a strict relationship between the kidney and the heart with different bidirectional and dynamic pathways, including the hemodynamic interactions in heart failure (HF) [2]. Cardiovascular damage/dysfunction can induce kidney injury and worsen kidney function by different mechanisms; in turn, kidney dysfunction may worsen cardiovascular function by affecting both the circulatory system and the heart [3]. Thus, patients affected by cardiovascular disease (CVD) often suffer also of chronic kidney disease (CKD) and vice versa. 

Several evidence has documented hemodynamic alterations, neurohormonal dysregulation, inflammatory activation, fibrosis, endothelial dysfunction and atherosclerosis in the development and progression of cardiac and renal diseases, generating a vicious circle with a mutual damage in both organs [4]. This represents a common basis for the interaction between heart and kidneys, leading to the decline of both cardiac and renal function in CRS. 

In more than 20% of hospitalized patients with acute HF a worsening of renal function has been detected whereas in chronic HF patients CKD is present in more than 50% of individuals [2,5]. Similarly, more than 50% of patients with CKD have up to a 20-fold increased risk of CVD [6]. In different other reports, including both acute and chronic HF patients, the prevalence of CKD was estimated around 50% and 40%, respectively, and acute kidney failure (AKI) ranged between 23% and 35% [7,8]. The coexistence of both kidney disease and CVD worsens their prognoses with an increased duration of hospitalization (+2 to 4 days) and an enhanced risk of rehospitalization and mortality during a follow-up of 6 months to 4 years [9]. Therefore, the identification of patients with CRS has a great implication in terms of prognosis since the coexistence of both heart and renal impairment may worsen the prognosis in these patients. Indeed, CRS is characterized by a high mortality rate and by a high in-hospital mortality rate, particularly in type 1 CRS [10].

A CRS diagnosis includes the concomitant presence of signs and symptoms of HF and the evidence of structural and functional kidney abnormalities. In this regard, biomarkers are invaluable tools to individually predict HF or renal disease, as well as in identifying cardiac dysfunction during renal diseases and renal impairment in HF. Biomarkers can have prognostic significance and provide insights into pathophysiology, and eventually can contribute to guide therapeutic approaches in CRS patients [11,12]. Furthermore, advancements in the pathophysiology of CRS contributed to the identification of novel biomarkers potentially useful in the diagnostic process and in the definition of appropriate therapies in CRS. Nonetheless, the therapeutic lines of CRS patients derive mainly from landmark HF or CKD trials, since CRS patients are not specifically studied in dedicated clinical trials, as well as they are not included or completely defined in the available HF and CKD trials. Hence, despite the improved recognition of CRS as a distinct disease with associated high morbidity and mortality, the pathophysiology and the specific therapeutic approaches of both acute and chronic CRS types remain incomplete and are still under investigation. Therefore, future focused trials are required to define the optimal management of CRS patients to improve prognosis and reduce mortality. 

It is worthy of note that pediatric CRS types are also currently described, which have peculiar aspects with respect to adult CRS types. However, the prevalence, risk factors and prognosis of pediatric CRS is less characterized. An extensive description of pediatric CRS has been recently reviewed elsewhere [13]. In this review article, we will discuss the current knowledge on the pathophysiology of CRS in adults, as well as the utility of biomarkers in cardiac and kidney dysfunction and potential insights into novel therapeutics for the management of CRS patients.

## 2. Classification of Cardiorenal Syndrome

CRS in adults is currently classified into five groups according to clinical presentations that highlight the importance of the underlying pathophysiology and the organ primary involved [14] to facilitate the early differential diagnosis and the most appropriate therapeutic path.

The five groups of CRS in adults are presented in Table 1: 

## 3. Pathophysiology of Cardiorenal Syndrome

Kidney function and heart performance are interconnected by the activation of systems that modulate bidirectional and dynamic pathways [14]. However, the exact pathophysiology of CRS remains still unclear, although few mechanisms have been extensively studied, including reduced cardiac output, increased central venous and abdominal pressure, increased oxidative stress, inflammation, and activation of the renin angiotensin aldosterone system (RAAS) [15] (Figure 1). 

### 3.1. Hemodynamic Factors

For many years the development of renal failure in presence of acute or chronic cardiac dysfunction has been attributed to the reduction of blood flow to the kidney (i.e., CRS types 1 and 2). Severe acute depression of cardiac function may lead to acute renal impairment as described in CRS type 1. The renal hypoperfusion in patients with HF may contribute to induce the activation of RAAS in juxtaglomerular kidney cells [16], as well as the sympathetic nervous systems activation and the increased secretion of vasopressin, resulting in fluid retention and preload increase, which further worsen cardiac function [17]. This may cause maladaptive hemodynamic effects in the kidneys, as well as ischemic effects in renal medulla [18]. If not adequately counteracted, the neurohormonal activation may be responsible for the vasoconstriction of the efferent arteriole and, consequently, also for the afferent arterioles and branches of the renal arteries, enhancing further the renal global hypoperfusion. Furthermore, the reduction of the blood supply to the kidney causes ischemia especially in the medullary area, whose metabolism is oxygen dependent, causing further ischemic damage to the organ [19]. The kidney, in turn, attempts to respond to the hemodynamic alterations with increased filtration fraction, which subsequently lead to hyalinosis of the glomeruli. This contributes to increased fluid and electrolyte reabsorption in the proximal tubules, leading to hypertrophy of the tubules and ultimately to the reduction of glomerular filtration rate (GFR) and tubular function [19]. In patients with a progressively reduced cardiac index, Ljungman et al. showed that GFR becomes dependent on afferent arteriolar flow in the most severe HF patients. Hence, in CHF patients renal blood flow and GFR are progressively reduced, suggesting the occurrence of a non-compensatory filtration fraction response despite the increased efferent arteriolar tone due to the counteractive stimulation of hormonal pathways [20]. The perpetuation of the damage and the constant reduction of renal function may induce electrolyte imbalance, increased uremia, the secretion of ouabain-like substances, acidosis and increased reactive oxygen species (ROS) production, which, in turn, are capable of leading to myocardial remodeling/dysfunction, microvascular dysfunction and relative cardiac ischemia [21]. In addition, other hemodynamic determinants of worsening renal and cardiac function may be attributed to the increased central venous pressure (CVP) and increased intra-abdominal pressure (IAP) as observed in right HF, and in HF with preserved or reduced ejection fraction [22,23]. Preclinical studies have shown that an elevated CVP plays a central role in renal dysfunction [24,25]. Clinical studies demonstrated that the worsening of renal function in patients with advanced decompensated HF was lower for those with low CVP (<8 mmHg) [22]. Thus, venous congestion has an important role in AKI development. It has been reported that in patients with acute decompensated HF the increased IAP have a strong correlation with renal function [26]. Moreover, in patients with CVD the increased CVP (>6 mmHg) may independently predict all-cause mortality in presence of impaired renal function [23]. Increased CVP may lead to a reduction in the glomerular filtration gradient and consequently reduced renal function [18]. In the Evaluation Study of Congestive Heart Failure and Pulmonary Artery Catheterization Effectiveness (ESCAPE) trial, the right atrial pressure was the only hemodynamic parameter that correlated with the increased serum creatinine [27]. The increased right ventricular afterload or the dysfunction of the right ventricle may increase CVP and renal venous pressure with negative consequences in glomerular filtration, contributing to worsen renal hemodynamics [28,29]. Nonetheless, different studies have shown that the incidence of renal failure is independent of blood pressure levels, and it is comparable among HF patients with reduced or preserved left ventricular ejection fraction (LVEF) [1]. Thus, it is reasonable to think that independently of the hemodynamic alterations, different other mechanisms are potentially involved in the pathophysiology of CRS as suggested by several evidence [1,30,31,32]. These mechanisms may include endothelial dysfunction; imbalance between ROS production/nitric oxide (NO) bioavailability; persistent RAAS activation; chronic inflammation (with neutrophil migration, leukocyte trafficking, cytokine production, cell apoptosis, chemokine secretion and immunologic imbalance) and the involvement of small noncoding RNAs and epigenetics alterations [1,27,28,29,30,31,32].

### 3.2. Endothelial Dysfunction, Oxidative Stress and Inflammation

Endothelial dysfunction has been shown to play an important role in the pathogenesis of CRS (particularly in chronic types 2 and 4 CRS) [33]. Endothelial dysfunction is characterized by reduced NO availability, increased oxidative stress and inflammatory processes, cell apoptosis, changes in the basal membrane structure and extracellular matrix composition and increased vascular permeability, as well as complement activation and stimulation of a prothrombotic state [34]. 

The reduced NO bioavailability causes an alteration in endothelium-dependent vasodilation resulting in a switch of vascular smooth muscle cell (VSMC) phenotype from the contractile to the synthetic state with a consequent increased cell proliferation and synthesis of extracellular matrix components, which lead to disorganization of the vascular cells lining [35]. Increased levels of angiotensin II and endothelin-1 activate vascular nicotinamide adenine dinucleotide phosphate (NADPH) oxidase, which is responsible of hyperproduction of ROS, leading to the activation of tissue inflammation and vascular remodeling, as well as to impaired responses to vasodilating agents [36,37]. 

An important determinant of endothelial dysfunction in CKD is the increased production of NO synthase inhibitor asymmetric dimethylarginine (ADMA), which contributes to renal damage by promoting renal ischemia, fibrosis and glomerular sclerosis [38]. On the other hand, endothelial dysfunction induced by ADMA contributes also to CVD development [39]. 

Innate and adaptive immune cell responses and professional and nonprofessional antigen-presenting cells are involved in the development of CRS. In type 1 CRS it has been demonstrated that the stimulation of toll-like receptors (TLRs) is linked to pathological apoptosis of endothelial and renal tubular epithelial cells with increased levels of proinflammatory cytokines, including caspase-8, interleukin-6 (IL-6) and tumor necrosis factor (TNF)-alpha [32]. Preclinical studies have also shown that activated endothelial cells may act as nonprofessional antigen-presenting cells, contributing to the pathogenesis of CRS [32].

Vascular endothelial growth factors (VEGFs) are important player in maintaining endothelium integrity and endothelial function by promoting endothelial cell survival and the synthesis of vasodilatory mediators. Different studies have demonstrated that a loss of the benefits derived from the VEGF-related cascade in physiological conditions may contribute to the pathophysiology of CRS [40]. VEGFA promotes cell survival and the synthesis of vasodilatory mediators through the modulation of phospholipase C (PLC)γ/protein kinase C (PKC) and phosphoinositide 3-kinase (PI3K)/Akt signaling [41]. In the presence of increased oxidative stress and inflammation, VEGFA becomes a mediator of further ROS accumulation and promotes the production of proinflammatory interleukins and profibrotic growth factors (including transforming growth factor β1 (TGFβ1), connective tissue growth factor (CTGF) and the metalloproteinases inhibitors) in endothelial cells, in podocytes, mesangial cells of glomeruli and tubules [42]. The switching of VEGFA effects is evident particularly in diabetes, contributing to endothelial dysfunction, inflammation and vascular disease [43].

The angiopoietins (Angpt) are vascular growth factors that may be involved in CRS pathophysiology. The two major isoforms of Angpt, Angpt1 and Angpt2, are pivotal in maintaining vascular homeostasis in opposite fashion. Through the Tie2 receptor, Angpt1 contributes to endothelial protection and microvascular vasodilation through NO release, as well as to the survival and maturation of endothelial cells and the phosphorylation of vascular endothelial adhesion molecule (VE)-cadherin [44]. On the other hand, Angpt2 acts as an inhibitor of Tie2 receptor resulting in the destabilization of the vessel wall, endothelial apoptosis, stimulation of inflammation, fibrosis, and hypoxia [43]. In these conditions, epithelial and endothelial cells differentiate towards a mesenchymal phenotype, losing their normal function and contributing to vascular remodeling and to the alterations of the vascular barrier. These alterations may result in the progression of cardiovascular and renal damage [45]. An independent role of Angpt2 in predicting adverse cardiorenal outcomes in patient with diabetes, coronary artery disease and HF has been described [46]. An increased Angpt2/Angpt1 ratio in the myocardium favors endothelium apoptosis and inflammation, which have been associated with the severity of renal impairment and albuminuria [47]. 

Increased oxidative stress is an important determinant of endothelial dysfunction and cardiovascular and renal damage [38]. Moreover, oxidative stress plays a central role for the signal transduction in cardiac cells in pathological conditions, including HF. ROS induce inflammatory cytokines production, decrease NO–cyclic guanosine monophosphate signaling, impair endothelial function and induce mitogen-activated protein (MAP) kinases, which are all involved in cardiac hypertrophy and remodeling, as well as myocardial dysfunction [48,49].

The increased activity of myocardial NADPH oxidase and the reduced function of superoxide dismutase in kidney may contribute to the increased ROS production in CRS [50], leading to myocardium and kidney damage [51]. 

Heart and kidney have high mitochondrial content, which represents one of the sources of ROS production under pathological conditions, leading to cardiorenal damage [52]. In a proinflammatory environment, mitochondrial fission may occur leading to cardiomyocyte apoptosis [53]. Acute kidney injury may induce mitochondrial fragmentation in heart tissue through the phosphorylation of dynamin-related protein 1 (Drp1), contributing to the cardiorenal damage [53]. 

Increased ROS production induces damage of DNA, lipids, proteins, and carbohydrates in heart and kidney [54]. ROS may activate fibroblast derived matrix metalloproteinases (MMP) [55], which are responsible of the extracellular matrix degradation, inflammation, and tissue remodeling in the heart [56]. These alterations are associated with the dysregulation and increased levels of beta-2-microglobulin and tissue inhibitor of metalloproteinases 1 (TIMP 1), which are linked to the severity of CRS [57]. It has been shown that apocynin, an inhibitor of NADPH oxidase exerts antioxidant properties, contributing to the protection of heart and kidneys with beneficial effects particularly in type 4 CRS. [58]. 

ROS contribute to tissue inflammation with increased levels of inflammatory mediators, including IL-6, C-reactive protein (CRP) and TNF, which are detected in both acute and chronic kidney disease and acute and chronic HF. The inflammatory processes in chronic cardiovascular and renal diseases are associated with increased risk of myocardial infarction and mortality [59]. In vivo experiments showed a significant increase of serum levels of proinflammatory cytokines in patients with type 1 CRS compared to those with acute HF without renal insufficiency [60]. 

Interestingly, recent clinical trials have shown that among drugs recently used as first line therapy in congestive HF with reduced ejection fraction, sodium-glucose cotransporter-2 (SGLT2) inhibitors may reduce oxidative stress with beneficial effects on cardiovascular and renal function [61]. 

In patients with CKD the intestinal microbiota is imbalanced in a condition of dysbiosis, which may enhance inflammatory mediators and contribute to the development of CRS. The bowel edema linked to HF congestion has been described as a potential contributing factor. It may cause the disruption of endothelial cells in intestinal villi with release of lipopolysaccharide (LPS) from gut bacteria and inflammatory cytokines such as IL-1, IL-6 and TNF-alpha [62]. Moreover, in the presence of impaired renal function, the colon becomes the organ deputed to urea excretion, leading to the growth of urease-positive species. These conditions are associated with the increased intestinal permeability and the activation of local and systemic inflammation. In addition, these types of bacteria increase the production of uremic toxins such as p-cresyl sulfate, indoxyl sulfate and trimethylamine N-oxide, which are characterized by profibrotic and proinflammatory effects [62]. 

### 3.3. Micro RNAs

An increasing body of evidence supports the role of micro RNAs (miRNAs) in pathophysiological conditions of heart and kidney. miRNAs are involved in different biological processes, including cellular differentiation and proliferation, apoptosis, inflammation, hemostasis, myocardial fibrosis, myocyte hypertrophy, cardiac remodeling and regeneration [63,64]. It has been shown that high levels of MiR-21 may be associated with cardiac remodeling and fibrosis of both the heart and kidneys [65,66] by several mechanisms, including the activation of collagen and alpha-smooth muscle actin (a-SMA) protein expression in myofibroblasts in the heart, as well as the inhibition of Notch2 expression in the kidney [67]. Preclinical findings suggest that suppression of miR-21 through the antisense oligonucleotide (ASO) may be considered as a promising possible therapeutic option in cardiorenal diseases [66,68]. 

## 4. Biomarkers in Cardiorenal Syndrome

In CRS biomarkers are important tools in assessing diagnosis, risk prediction, and prognosis in patients with HF and impaired kidney function, also providing insight into pathogenetic pathways and therapy in CRS patients [69,70]. Some biomarkers reflect hemodynamic changes, as well as heart and kidney damage and/or dysfunction, and others are the expression of changes in collagen turnovers in the extracellular matrix of both heart and kidneys, while others may reflect oxidative stress-induced cell damage (Table 2). 

Combining cardiac and renal biomarkers in multimarker strategies is a promising approach in CRS management since may increase the accuracy of individual biomarkers for the diagnosis, therapy, and prognosis of CRS [1,71,72]. Nevertheless, the selection and appropriate application of biomarkers, as well as their optimal combinations, remain a complex process that must be interpreted within different clinical situations [73]. 

### 4.1. Cardiac Biomarkers

Among specific cardiac biomarkers, cardiac troponins (cTns) (including cTnT and cTnI) raise in CRS and correlate with ventricular remodeling in HF [74]. In patients with advanced HF on optimal medical therapy, increased cTns levels are associated with clinical congestion, as detected by peripheral edema and pulmonary rales, high pulmonary arterial pressure and high pulmonary capillary wedge pressures [75]. However, their clinical applicability as a surrogate of congestion is limited by the lack of evidence of the correlation of changes over time of cTns levels with clinical congestion. cTns predict prognosis and may stratify the cardiovascular risk in HF patients [76]. The reduced filtration rate in patients with CKD is also associated with elevated cTns levels, which predict cardiovascular and all-cause mortality in patients with all stages CKD [77]. 

Natriuretic peptides (NPs) (including B-type natriuretic peptide (BNP), N-terminal pro B-type natriuretic peptide (NT-proBNP)) are the most used and recognized biomarkers in chronic and acute HF and are also raised in patients with CKD [78]. The assessment of plasma levels of NPs has been demonstrated to predict cardiovascular risk in patients with CKD [79]. Moreover, an elevated NT-proBNP/BNP ratio has been detected in acute HF patients with impaired renal function with prognostic significance for CRS development [80]. Plasma NPs concentrations have been shown to rise in parallel with decreasing renal function and are associated with an increased risk of progression to end-stage renal disease (ESRD) [81]. It has been shown that the combination of an elevated NT-proBNP and reduced eGFR (<60 mL/min) is a predictor of 60-day mortality in acute HF patients [65]. The reduction of NT-proBNP levels during therapeutic treatment of acute HF patients has been associated with improved outcomes, even in patients with reduced renal function [82]. 

Nevertheless, the cut-offs of both cTn and NPs in a setting of CRS, and particularly in relation to age, gender and stage of renal insufficiency, are not univocally established. In such a context, specific studies should be planned [83].

In addition to these traditional cardiac biomarkers, different other biomarkers have been identified for the diagnosis and prognosis of CRS. Galectin-3 is a component of the beta-galactosidase-binding lectin family, which is released by activated macrophages. Galectine-3 induces the deposition of collagen in the extracellular matrix promoting fibrosis at renal and cardiac level. Patients with elevated Galectin-3 levels showed an accelerated decline of GFR [84,85]. 

VEGF, platelet-derived growth factor (PDGF) and soluble VEGF receptor-1 (sFlt-1) are elevated in patients with HF. VEGF exerts endothelium protective effects by promoting endothelial cell survival and increasing NO and prostaglandin release from endothelial cells. PDGF contributes to infarct size reduction and to the improvement of cardiac dysfunction. On the other hand, sFlt-1 may inhibit the protective roles of PDGF contributing to the endothelial dysfunction and microvascular alterations in patients with HF and CKD [86]. PDGF/sFlt-1 ratio correlates with HF severity in patients with renal dysfunction, and therefore, sFlt-1 could be considered as a biomarker with prognostic significance in HF patients [87]. 

Soluble Suppressor of Tumorigenicity-2 (sST2) is a member of the IL-1 receptor family, which affects the production of T-helper type 2 (Th2)-related cytokines [88]. sST2 correlates with CV events and mortality in patients with acute and chronic HF [89,90], as well as with the development of CKD. Moreover, sST2 is associated with the risk of CV events and HF development in patients with renal dysfunction [91,92]. 

Myeloperoxidase (MPO) is produced by neutrophils and monocytes and is related to oxidative stress, inflammation, ventricular remodeling and vulnerable atherosclerotic plaque [93]. MPO is elevated in CHF patients and positively correlates with CHF progression [94], and it has an independent prognostic value in patients with CHF and AHF as well [94]. Moreover, MPO may predict CKD severity and mortality in patients with CKD [95]. 

Procalcitonin is increased in the context of inflammation, particularly during bacterial infection, and identifies patients at increased risk of AKI [96]. Patients with HF have high plasma procalcitonin levels independently of concomitant evidence of infection, since inflammation is a landmark in the pathophysiology of HF with prognostic significance. Procalcitonin levels may predict HF severity along with mortality and rehospitalization for decompensated HF. Thus, procalcitonin is increasingly considered a prognostic marker in HF also providing information on therapeutic responses [97]. 

Copeptin is a C-terminal portion of arginine vasopressin (AVP), which causes arteriolar vasoconstriction, increased vascular resistance and water reabsorption in the distal tubule of the kidney. It is considered a marker of activated hypothalamus pituitary-adrenals axis. There is evidence that copeptin could be associated with CVD in patients with CKD [98], as well as it could be considered as a marker of AHF and AKI.

Other promising biomarkers include plasma thiobarbituric acid-reactive substances, 8-epi-isoprostanes, soluble thrombomodulin and Angpt2, which are elevated in patients with CVD who develop kidney injury [67]. Moreover, mid-regional proadrenomedullin (MR-proADM) and liver-type fatty acid-binding protein (L-FABP) have emerged as biomarkers able to predict the decline of renal function and morbidity in patients with HF [99]. Urinary cofilin-1 is a modulator of epithelial–mesenchymal transition, which has attracted attention as an intriguing biomarker in CRS, since it is related to the severity of HF in patients with acute renal failure [67]. 

### 4.2. Renal Biomarkers

Among biomarkers of renal function serum creatinine levels and GFR are largely used to identify renal impairment and prognosis in patients with renal diseases and are potentially useful in identifying the increased prevalence of renal dysfunction in patients with HF. Nevertheless, serum creatinine levels may not accurately reflect GFR, since it could be influenced by nonrenal factors such as sarcopenia, which is detected in 20% of CHF patients [100]. In this regard, other more accurate biomarkers of renal dysfunction have been identified. Cystatin C, a cysteine proteinase inhibitor, is filtered through the glomerulus and then reabsorbed by tubular cells. It has been described as a more accurate surrogate marker of GFR compared to serum creatinine levels, since it is less dependent on age, nutritional status and body mass index [101]. Cystatin C may stratify the risk of CV events, including coronary artery disease, and acute and chronic HF. A rise in cystatin C levels in acute HF predicts short-term prognosis with higher in-hospital mortality and longer duration of hospitalization. Moreover, it may increase the accuracy of NT-proBNP in type 1 CRS. Nevertheless, caution should be used on its routinary clinical use, since the cystatin C levels may increase in other clinical conditions, including thyroid dysfunction, obesity, inflammatory diseases and corticosteroid therapy [102]. Thus, according to the KDIGO guidelines, the use of Cystatin C is recommended when the presence of renal dysfunction cannot be confirmed by serum creatinine levels alone [103]. 

Under some circumstances, including elevated glomerular pressure, tissue inflammation and endothelial dysfunction, the damage of the glomerular membrane may cause an increased excretion of albumin, which reflects the anomalous renal microcirculation. Both microalbuminuria (urinary albumin/creatinine ratio, UACR, 30–300 mg/day) and macroalbuminuria have been associated with increased risk of HF, independently of hypertension and diabetes, probably as the result of generalized endothelial inflammation. In HF patients, microalbuminuria may be associated with altered renal hemodynamic and may better identify the stage of CKD and stratify the prognosis [104,105]. 

Interleukin-18 (IL-18) induces T-lymphocytes and natural killer cells to produce interferons [106]. It has been shown that IL-18 is involved in renal damage related to apoptosis and ischemia-reperfusion injury and may predict cardiovascular prognosis [107,108]. Furthermore, it may identify acute renal insufficiency at early stages, as well as it may predict prognosis in AKI [71]. The IL-18 levels correlate with vascular stiffness and predicts mortality in CKD patients, and it may promote HF progression by inducing inflammation, cell necrosis and myocardial ischemia [109]. The IL-18 levels are increased in HF patients and are related to the reduced LVEF and correlate to the increased mortality in HF patients. Nevertheless, further studies are needed to determine the utility of routinary use IL-18 in CRS [60].

The role of tubular biomarkers in predicting the progression of renal disease has been investigated particularly in patients with HF. Tubular biomarkers may be potentially useful to identify patients at high risk for CRS, as well as to establish prognosis and assess the optimal response to therapies. Among them, N-acetyl beta glucosaminidase (NAG) is a lysosomal protein detected in urine after tubular damage. It has been shown that NAG levels correlate to cardiac or renal dysfunction, and its levels are also found in patients with urinary infection. NAG may predict prognosis in patients with AKI, CKD, or HF [110]. 

Kidney injury molecule 1 (KIM1) is a transmembrane glycoprotein expressed in proximal tubule cells after hypoxic injury and may identify the development of AKI or CKD in patients with HF. Moreover, KIM1 is associated with HF, cardiovascular events and deaths in patients with AKI and CKD [111]. Within 24–48 h after kidney injury, KIM-1 expression is considerably increased in proximal tubular epithelial cells. Its levels correlate to blood creatinine peak, multiorgan system failure, and oliguria. Moreover, urinary KIM1 has been demonstrated to predict death, myocardial infarction and HF hospitalization [111]. 

Alpha-1 microglobulin (A1M) is filtered by glomerulus and is completely reabsorbed by the renal tubule. Its urine levels increase during renal tubules damage. Increased concentrations of beta-2 microglobulin (B2M) have been also detected when the renal tubules are damaged [112]. Hence, these molecules can function as potential markers of kidney damage, although their role must be further determined.

Neutrophil gelatinase associated lipocalin (NGAL) is a small protein filtered through the glomerulus and reabsorbed in the proximal part of the tubule. It is an early marker of tubular damage and worsening of renal function, as well as correlates with residual renal function in patients in dialysis. Within hours after acute renal injury, NGAL messenger RNA is transcribed in the tubule cells [113], and its levels are detected 24 h before the rise of creatinine. It is a useful marker for adverse clinical outcome in patients with AKI and CKD [114]. Urinary levels of NGAL are significantly higher in HF patients and independently correlate with GFR, urinary albumin excretion and NT-proBNP [115], and functions as a biomarker to predict mortality rate in CHF patients [116]. NGAL also correlates with inflammatory mediators and ventricular remodeling in AHF [117]. However, NGAL evaluation loses specificity in the presence of conditions such as sepsis, inflammation, anemia and hypoxemia. 

Liver-fatty acid-binding proteins (L-FABP) belong to the family of tissue-specific FABPs that is expressed in tubular epithelial cells and is excreted into urine with cytotoxic lipids. It has been shown that urinary L-FABP is associated with ischemic tubular injury and with the risk for acute kidney failure in type 1 CRS [118]. 

Tissue inhibitor of metalloproteinase 2 (TIMP-2) and insulin-like growth factor-binding protein 7 (IGFBP7) levels are increased in conditions of inflammation, ischemia, and oxidative stress. Both have been demonstrated to be able to predict the risk of acute kidney injury during hospitalizations in intensive care unit with a greater accuracy compared to KIM-1, NGAL, L-FABP [119].

Proenkephalin A (PENK) is a precursor and surrogate marker of enkephalin. It acts on opioid receptors, which are widely distributed with the highest densities in the kidney. It has been shown that PENK levels may reflect cardiorenal status in acute HF [120,121], functioning also as prognostic marker for renal function worsening and it is an independent predictor of poor renal outcomes [122]. Moreover, PENK may predict in-hospital mortality as well as major adverse cardiac events, including death, reinfarction and rehospitalization for HF [123]. Thus, PENK could be a promising novel predictive and prognostic marker of early diagnosis of type 1 CRS [121,122]. 

Urinary angiotensinogen (uAGT) is also increased in AKI, and it has been also associated with the progression of kidney disease, probably reflecting the local activation of RAAS within the kidney [124,125]. Thus, it attracted attention as an emerging biomarker in CRS.

Undoubtedly, individual biomarkers may help in the diagnosis, prognosis, and therapeutic assessment of patients with HF, kidney impairment and CRS. Nevertheless, they must be interpreted within different clinical pictures. Although there is no conclusive evidence about the use of individual biomarkers for the diagnostic and prognostic assessment in CRS patients, combination of biomarkers with the best characteristics in terms of specificity and sensitivity might represent the best future diagnostic and prognostic strategy. 

## 5. Therapeutic Strategies in CRS

The management of CRS patients is a real challenge, considering the complex and heterogeneous pathophysiology of CRS. Furthermore, each patient has his own personal history and risk profile due to combination of comorbidities. It is worth of note that the main causes of CV death in patients with kidney diseases are specific types of cardiomyopathies, atherosclerosis, and CHF related complications. Additionally, the medications used for the treatment of CRS are not fully studied in randomized clinical trials specifically designed for CRS. Hence, there is not a complete agreement on therapeutic strategies that could be recommended for CRS patients, rather current therapeutic options derive mainly from sparse and not complete evidence from HF studies and/or small preclinical and clinical studies [1,4,10,15]. Drugs that can slow down the decline of renal function are of utmost importance, since kidney dysfunction in the setting of HF has a strong prognostic relevance. In this regard, large benefit derives from drugs that improve renal flow and function in different types of CRS. Therefore, many drugs used in CRS patients are drugs commonly used in CHF patients with or without renal dysfunction [126]. Thus, in this section we will discuss the major lines of evidence regarding the main drugs used in CHF that may exert benefit also in CRS patients. Along with diuretics/ultrafiltration and inotropic agents, other drugs used in HF with reduced ejection fraction (HFrEF) with class I level of evidence includes beta blockers (BB), Angiotensin Converting Enzyme inhibitors (ACEI)/Angiotensin Receptor Blockers (ARBs) or Angiotensin Receptors Neprilysin Inhibitors (ARNI), Mineralocorticoid Receptor Antagonists (MRA), and Sodium Glucose Transporter inhibitors (SGlT2i). These drugs are known to control intravascular volume and modulate neurohormonal activation in HF patients [94]. Therefore, they may exert beneficial effects also on kidney function in CRS (particularly in Types 2 and 4 CRS). Other treatment options include implantable defibrillator therapy (ICD) and cardiac resynchronization therapy (CRT) [126] (Table 3).

### 5.1. Diuretic and Ultrafiltration Therapy

In patients with acute or chronic HF, central and peripheral congestion is commonly detected, and diuretics represent an important therapeutic tool with or without CRS. 

However, although diuretics improve HF symptoms, they have no beneficial effects on HF hospitalizations and mortality [95]. Loop diuretics (including furosemide, bumetanide, torsemide and ethacrynic acid) are the diuretics of choice in acute or chronic HF [126,127]. In acute decompensated HF patients, the addition of acetazolamide to loop diuretics improves the diuretic efficiency in terms of successful decongestion. However, in the Acetazolamide in Decompensated Heart Failure with Volume Overload (ADVOR) trial acetazolamide treatment did not show any effect on mortality, kidney function or hypokalemia, even though it was well tolerated [128]. Although diuretic synergy could be useful in some context in acute HF patients, it is not fully assessed whether this concept could be transferred also to CRS. The use of diuretics may induce worsening of renal function particularly in patients with advanced HF [127]. A proposed mechanism is the potential RAAS activation during treatment with high-doses of diuretics, although different studies (DOSE-AHF and CARRESS-HF) did not show significant differences in RAAS activation between treatments with high-dose and low-dose of loop diuretic [129,130].

CRS patients may often develop diuretic resistance, which is associated with renal impairment, increased risk of rehospitalization and mortality in HF patients [131]. This condition consists in the attenuation of the maximal diuretic effect that limits sodium and chloride excretion during diuretic use [132]. In CRS, different factors are related to diuretic resistance, including the decreased tubular excretion of diuretics and the increased proximal reabsorption of sodium due to RAAS activation [133]. Hypochloremia may play a critical role in neurohormonal activation in patients with HF on high dose of loop diuretics, which may contribute to diuretic resistance in these subjects [134].

Diuretic use can induce the braking phenomenon, consisting in the diminished diuretic efficacy with each successive dose and the induction of distal tubular hypertrophy in the long term [127]. Sodium repletion can attenuate the braking phenomenon [135]. It has been proposed that enhanced distal sodium transport may attenuate the maximal efficacy of furosemide. The combined use of different types of diuretics may increase their efficacy. It has been suggested that the combined use of thiazide-type diuretics may increase furosemide-induced sodium excretion [136]. Diuretic efficiency (defined as the fluid lost per milligram of loop diuretic in acute HF patients) is considered a prognostic marker in CRS. Patients with reduced diuretic efficiency have an increased risk of death, HF rehospitalization compared with those with normal efficiency [137] and are more likely to experience worsening of renal function [138]. 

Ultrafiltration is a mechanical process that removes isotonic liquid and low molecular weight molecules from the circulatory system, eliminating the liquid excesses without neurohormonal activation. Ultrafiltration is useful in patients with severe HF with fluid retention and resistance to diuretic treatment [139,140]. Different studies suggested the efficacy of ultrafiltration in CRS patients. In the RAPID-CHF study better results were found in CRS patients where ultrafiltration was used, rather than a classic drug treatment [141]. In the UNLOAD trial, patients with acute HF who underwent ultrafiltration lost more weight than those on diuretic treatment, with lower readmittance rate after 90 days after hospital discharge, even though no improvement of renal function was observed [142]. Moreover, in patients with acute HF, ultrafiltration treatment was not associated with renal function worsening [143]. 

### 5.2. Inotropic Agents and Beta Blockers

In the setting of type 1 CRS, the use of inotropes may contribute to improve cardiac output and reduce venous congestion [144]. Among inotropes, dopamine induces cardiac inotropic effect, systemic vasoconstriction and improves renal blood flow through its effects on the β- and α-adrenergic receptors, as well as the renal dopaminergic receptors [145]. Although some studies suggested a renal protective effect of low-dose dopamine in acute HF, no long-term benefits were demonstrated [146]. Moreover, low-dose dopamine improved urine output without effects on rehospitalization and mortality [147]. Few and sparse data are available about the use of other inotropes in CRS [1]. 

BB are included in the first-line therapy of chronic HF, with evidence of the striking improvement of HF prognosis [126]. However, BB are not suggested as the treatment option for patients with acute decompensated HF and in CKD patients without HF [148], as well as no direct benefit has been proven in CRS. 

### 5.3. Renin Angiotensin System Inhibitors

Several clinical trials demonstrated that neuro-hormonal modulation in HF may contribute to reduce HF symptoms, reverse cardiac remodeling, and improve survival. Hence drugs that modulate neuro-hormonal activation in HF are becoming the pillar of the modern pharmacological approach in HF treatment [149]. Agents interacting with the neurohormonal systems might have a relevant role also in the therapeutic management of CRS (type 2 CRS in particular), since CRS is characterized by the dysregulation of neurohormonal responses, including RAAS hyperactivity that is closely connected to oxidative stress, inflammation and vascular remodeling [56,150,151]. In HF patients, RAAS inhibitors (i.e., ACEI, ARB or ARNI and MRA) have been shown to improve prognosis [126], with beneficial effects also on renal function. Although recent studies have demonstrated that RAAS inhibitors are safe in patients with advanced CKD and that they may protect from pathological hyperfiltration by improving intrarenal hemodynamic [152], controversial findings, mostly derived from small observational studies, raised a warning on the use of RAAS inhibitors particularly in CRS patients. There is evidence that RAAS inhibitors may potentially compromise the residual kidney function and accelerate GFR reduction in patients with CKD [153]. Nevertheless, ACEi do not slow the decline of GFR in HFrEF [154]. ARNI have shown benefits for HF outcomes, in both chronic and acute HF [155,156]. ARNI can preserve renal function more effectively than ACEi and ARB by inhibiting the progressive decline of GFR associated with HF [157,158,159,160], even though data in patients with eGFR < 30 mL/min are lacking [161]. As a result of these conflicting reports, physicians are frequently reluctant about the use of RAAS inhibitors, including ARNI in patients with advanced chronic CKD, even when these compounds could be recommended for other concomitant indications such as HF [126]. This may frequently cause the decrease of the dosages or even the interruption of these treatments. Therefore, specific studied focusing mainly on CRS patients are required.

### 5.4. SGLT2 Inhibitors: An Emerging Therapeutic Tool in CRS

SGLT2i were originally used as antidiabetic drugs, and early clinical trials, including CANVAS [162], DECLARE-TIMI [163] and EMPA-REG OUTCOME [164], have demonstrated their efficacy in reducing cardiovascular mortality and HF hospitalizations in diabetic patients with HFrEF. In particular, in these early trials designed for patients with type 2 diabetes mellitus (T2DM), SGLT2i showed better cardiovascular and renal outcomes, including a reduction of cardiovascular death; nonfatal MI; nonfatal stroke; HF hospitalizations and worsening of nephropathy (progression to macroalbuminuria, doubling of serum creatinine, ESRD or death for renal disease) [162,163,164]. In the CVD-REAL study (Comparative Effectiveness of Cardiovascular Outcomes in New Users of SGLT-2 Inhibitors) conducted on 309,056 patients, the benefits of SGLT2i in terms of reduction of HF hospitalizations and death were significantly higher compared to other glucose-lowering agents after the propensity matching analysis [165]. Subsequent studies demonstrated that patients treated with SGLT2i had a reduction of HF hospitalizations and CV death and a lower annual decline in renal function in a subset of HF patients with reduced and preserved EF [166,167,168], independently from diabetes and concomitant CKD. Furthermore, recent evidence confirms the efficacy of SGLT2i in terms of nephroprotection by reducing the decline of kidney function and CV mortality also in patients with CKD, regardless of the presence of diabetes [169,170]. A meta-analysis demonstrated a protective effect of SGLT2i also in acute kidney failure mainly because this class of drugs may improve tubulointerstitial hypoxia, maintain tubular cell integrity and prevent proteinuria [171]. Due to these clinical characteristics, SGLT2i might represent a promising therapeutic tool in the treatment of CRS. 

SGLT2is have an excellent diuretic and metabolic effect, as well as they may exert several other mechanisms, including neurohormonal modulation and reducing oxidative stress, inflammation and cardiovascular remodeling [172] (Figure 2). Experimental and clinical studies have demonstrated an excellent nephroprotective effect for this class of drugs, even stronger than that shown by ACEi or ARBs, which are considered the most effective drugs for preserving kidney function in HF patients [173,174]. By inhibiting Na^+^/glucose cotransporter 2, SGLT2i improves glycemic control and reduces intravascular volume, as well as reduces intraglomerular pressure by contrasting tubuloglomerular feedback with a consequent protective effect on the glomerular endothelium [175]. SGLT2i treatment maintains sodium delivery to the macula densa by attenuating glucose and sodium reabsorption, resulting in improved hemodynamic effects [169]. Thus, SGLT2i can cause natriuresis in the early phase of treatment, which may activate systemic RAAS. Nonetheless, RAAS activity seems to be not affected by chronic SGLT2i administration [176]. Furthermore, SGLT2i treatment has been demonstrated to reduce hyperfiltration by increasing urinary adenosine and prostaglandin concentrations without increasing the renal vascular tone [177]. 

In animal models, SGLT2is have shown to reduce renal damage during ischemia–reperfusion through the inhibition of cell apoptosis in the tubule by increasing hypoxia induced factor-1 (HIF-1) and by restoring the expression of VEGFA and improving endothelial rarefaction of peritubular capillaries [178]. SGLT2i may increase the systemic oxygen availability by improving tubulointerstitial hypoxia, allowing fibroblasts to resume normal erythropoietin production and inducing the suppression of sympathetic hyperactivity, allowing cardiovascular and renal protection [179]. Moreover, the shift of metabolism from glucose and fat to ketone bodies has been hypothesized as one of the mechanisms to lower renal oxygen consumption to alleviate hypoxic stress, as well as to improve renal function and slow the progression of diabetic kidney disease [180]. Interestingly, SGLT2is activate sirtuin-1, which exert protective actions on the heart and kidneys [181] by reducing fibrosis and inflammation in renal cells and reducing hypoxic injury in cardiomyocytes. SGLT2is can also inhibit the renal reabsorption of sodium coupled-uric acid inducing the urinary excretion of uric acid. This mechanism may further contribute to slow the progression of CKD and CVD [182]. SGLT2is have been also shown to stabilize the circadian rhythms of blood pressure and the sympathetic system activation by the interaction with neural signals within the hypothalamus [183]. 

### 5.5. Novel Therapeutic Strategies

Other therapeutic options have been proposed in CRS patients. Selective antagonists of the V2 receptor of arginine vasopressin have been tested in the treatment of HF with controversial results. In the EVEREST program (Efficacy of Vasopressin Antagonist in Heart Failure Outcome Study With Tolvaptan), the V2 receptor antagonist tolvaptan did not achieve benefits in term of reduction of cardiovascular death and HF hospitalizations in patients with acute HF and LVEF < 40% [176]. Moreover, in the TACTICS-HF study (Targeting Acute Congestion With Tolvaptan in Congestive Heart Failure) [184] and in the SECRET CHF (Short Term Clinical Effects of Tolvaptan in Patients Hospitalized for Worsening Heart Failure With Challenging Volume Management), the addition of tolvaptan to furosemide did not improve the 24-h response to the diuretic treatment and dyspnea, respectively [185]. 

More recently, in preliminary studies in patients with HF, CKD and anemia erythropoietin receptor activation in the heart is generating increasing interest for future therapeutic strategies, since its activation may exert protective role against apoptosis, fibrosis and inflammation, leading to improved cardiac structure and function [186]. 

### 5.6. Non-Pharmacological Approaches

Finally, non-pharmacological approaches may also potentially impact on prognosis in patients with HF and CKD in terms of improvement of the cardiorenal status and mortality [1,126], although controversial issues exist. Several evidence has suggested that implantable cardiac defibrillators (ICD) are useful not only in HF patients but also in patients with CRS. In these patients ICD is recommended to reduce the risk of sudden death and all-cause mortality in those who have recovered from a ventricular arrhythmia associated with hemodynamic instability and in patients with symptomatic HF and a LVEF < 35% despite optimized medical treatment over at least 3 months. Furthermore, cardiac resynchronization therapy (CRT) is recommended if QRS duration on electrocardiogram is >150 ms, particularly if a left bundle branch block is present [126]. Although the concomitant presence of HF in patients with ESRD increases the global death prevalence of 50%, the use of device therapy is estimated to be very low in this category of patients (in less than 10% of cases) [187]. This is mainly due to the lack of evidence from randomized clinical trials. In particular a metanalysis from ICD trials has indicated no clear benefit from device implantation in CHF patients with reduced renal function [188], taking also into account the non-arrhythmic causes of death in these patients, as well as the high burden of non-cardiovascular comorbidities, including vascular access, bacteremia, bleeding and higher rates of lead related complications, which are factors that may reduce the net benefit of ICD implantation in in CRS patients [189]. 

## 6. Conclusions

CRS received much attention in preclinical and clinical studies, since it represents a serious healthcare problem with high morbidity and mortality. A growing body of evidence fostered the understanding of the strict relation between heart and kidney by showing different pathophysiological mechanisms involving various cell mediated signaling pathways. Nonetheless, the pathophysiology of acute and chronic CRS types remains incomplete and still under investigation.

In the last few years, different biomarkers have been investigated with the aim to achieve an early and accurate diagnosis of CRS, to provide a prognostic role and to guide the development of targeted pharmacological and non-pharmacological therapies. Biomarkers are useful in identifying cardiac dysfunction in renal diseases and renal injury in HF. Nevertheless, the use of the current available biomarkers in CRS is limited because of the paucity of evidence, which makes their classification and utility in different types of CRS difficult. A multimarker strategies combining cardiac and renal biomarkers is a promising approach in CRS management, since it may increase the accuracy of individual biomarkers for the diagnosis, prognosis and therapy of CRS patients. However optimal biomarkers combinations need to be further defined by specific studies. Indeed, CRS patients are currently underrepresented in distinct HF and CKD trials, and thus, further specific studies are required to assess the diagnostic paths, as well as to individualize specific therapeutic strategies. 

Novel drugs for HF are emerging as promising tools also in CRS. Several randomized controlled trials revealed considerable benefits with SGLT2i treatment in HF and CKD, regardless of the presence of T2DM. Thus, SGLT2is might represent a promising strategy in the management of CRS due to their ability to counteract the development of CRS through different mechanisms, including the restoration of tubuloglomerular feedback and the correction of tubular hypoxia and sympathetic overdrive. SGLT2is have shown to slow the progression of cardiac and renal dysfunction and may possibly improve the prognosis of CRS patients. Nevertheless, furthers studies in CRS patients should be performed to address specific aim on adequate therapies tailored on different types of CRS.

## Figures and Tables

**Figure 1 ijms-24-05089-f001:**
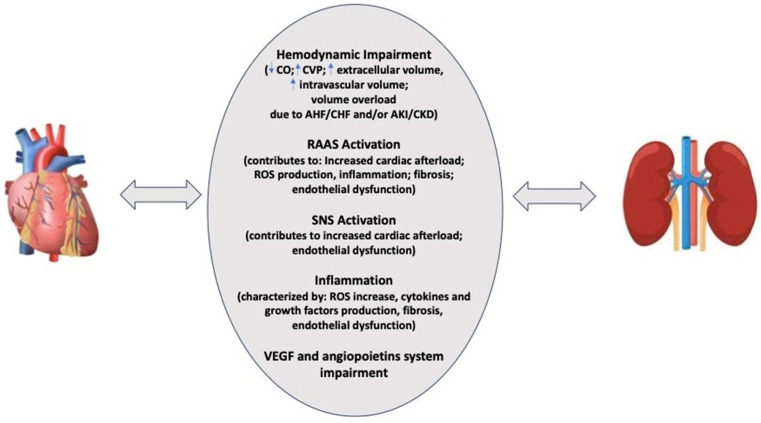
Main pathophysiological mechanisms of heart and kidney damage in cardiorenal syndrome. ↓ reduced; ↑ increased; AHF, Acute Heart Failure; AKI, Acute Kidney Injury; CHF, chronic heart failure; CKD, Chronic Kidney Disease; CO, Cardiac Output; CVP, Central Venous Pressure; RAAS, Renin Angiotensin Aldosterone System; ROS, Reactive Oxygen Species; SNS, Sympathetic Nervous System; VEGF, Vascular Endothelial Growth Factor.

**Figure 2 ijms-24-05089-f002:**
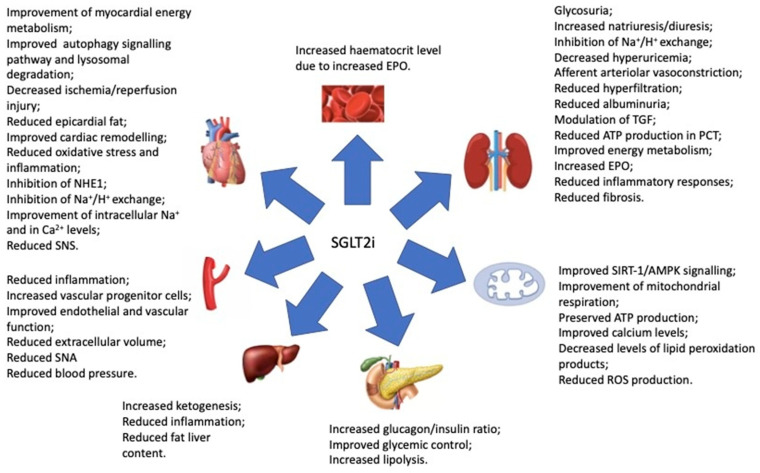
Systemic beneficial effects of SGLT2 inhibitors that contributes to cardiovascular and renal protection. AMPK, Adenosine Monophosphate Kinase; ATP, Adenosine Triphosphate; EPO, Erythrpoietin; NHE1, Sodium Hydrogen Exchange 1; PCT Proximal Convoluted Tubule; ROS, Reactive Oxygen Species; SGLT2i, Sodium Glucose Transporter 2 inhibitors; SIRT-1 Sirtuin 1; SNS, sympathetic nervous system; TGF, Tubuloglomerular Feedback.

**Table 1 ijms-24-05089-t001:** Classification of cardiorenal syndrome.

Definition	Description	Clinical Conditions	Main Pathophysiological Mechanisms of Heart and Kidney Damage
Type 1 CRS (Acute CRS)	Acute HF resulting in AKI	AKI in the setting of acute HF or cardiogenic shock	Venous congestion, Renal hypoperfusion, SNS/RAAS activation, Oxidative stress, Inflammation.
Type 2 CRS (Chronic CRS)	Chronic HF resulting in CKD	CKD in the setting of chronic HF The diagnosis of CKD is based on the Improving Global Outcomes Kidney Disease Outcomes (KDIGO) Quality Initiative (KDOQI) criteria: - albuminuria and/or glomerular filtration rate (GFR) < 60 mL/in/1.73 m^2^; - sustained decrease in GFR > 5 mL/min/1.73 m^2^/year; - decline of GFR > 10 mL/min/1.73 m in 2/5 years; - sustained increase in albuminuria along with suspected diagnosis of congestive HF before the onset or progression of CKD.	SNS/RAAS activation, Fibrosis, Oxidative stress, Inflammation.
Type 3 CRS (Acute renocardiac syndrome)	AKI resulting in acute HF	Acute HF in the setting of AKI. Acute HF is linked to acute worsening of renal function with consequent electrolyte imbalance, metabolic acidosis, and volume overload.	Volume overload, SNS/RAAS activation, Oxidative stress/ mitochondrial dysfunction, Inflammation, Electrolyte disorders and metabolic disorders (due to uremic condition).
Type 4 CRS (Chronic renocardiac syndrome)	CKD resulting in chronic HF	LVH/dysfunction and chronic HF in the setting of CKD (Uremic Cardiomyopathy) CKD is related to accelerated atherosclerosis, insulin resistance, lipid dysmetabolism, neurohormonal imbalance and consequently to the development of CVD.	SNS/RAAS activation, Inflammation/fibrosis Hyperphosphatemia, Secondary hyperparathyroidism, Increased levels of circulating erythropoiesis inhibitors, furans, phenols, beta-2-microglobulin, leptin and polyols.
Type 5 CRS (Secondary CRS)	Systemic processes resulting in both HF and kidney damage	Amyloidosis, Autoimmune Diseases (SLE), Sepsis, COVID-19, Advanced liver diseases, Hepatorenal syndrome, Cirrhosis.	Inflammatory and prothrombotic states, Secretion of proinflammatory Cytokines, Endothelial dysfunction, Impaired coronary and glomerular autoregulation.

AKI, acute kidney injury; CKD, chronic kidney disease; CRS, cardiorenal syndrome; HF, heart failure; LVH, left ventricular hypertrophy; RAAS, Renin Angiotensin Aldosterone System; SLE, Systemic Lupus Erythematosus; SNS, sympathetic nervous system [1,4,9,10,11,12,13,14,15].

**Table 2 ijms-24-05089-t002:** Biomarkers in Cardiorenal Syndrome.

Biomarkers	Characteristics	Clinical Utility
**Cardiac Biomarkers**		**CRS Type**
cTn	Marker of myocardial injury that correlates with ventricular remodeling in HF.	1, 2
Natriuretic peptides	Markers of myocardial increased wall stretch. They are the most used and recognized biomarkers in chronic and acute HF and are also raised in patients with CKD.	1, 2, 3, 4, 5
sST2	sST2 is a member of the IL-1 receptor family that affects the activation of Th2 cells and the production of Th2-related cytokines. sST2 correlates with cardiovascular events and mortality in patients with AHF and CHF. Moreover, sST2 correlates with the development of CKD, as well as to the risk of CV events and HF development in patients with renal dysfunction.	2, 5
Galectin-3	It is a component of the beta-galactosidase-binding lectin family, it is released by activated macrophages and induces the activation and deposition of collagen in the extracellular matrix promoting fibrosis at renal and cardiac level. Patients with elevated Galectin-3 levels showed an accelerated decline of GFR.	2, 4, 5
VEGF	Involved in the regulation of endothelial function and angiopoiesis; it may affect myocardial afterload. It is elevated in patients with HF.	2, 4, 5
PDGF	Involved in regulation of myocardial and kidney fibrosis. It is elevated in patients with HF.	2, 4, 5
sFlt-1	Soluble VEGF receptor associated with microvascular disease and impaired angiopoiesis. It is elevated in patients with HF.	2, 4, 5
Copeptin	It is considered a marker of activated hypothalamus pituitary-adrenals axis. There is evidence that copeptin could be associated with CVD in patients with CKD, as well as it could be considered as a marker of AHF and AKI.	1, 2, 3 4, 5
MR-proadrenomedullin	Involved in regulation of vascular leakage; it predicts the decline of renal function and morbidity in patients with HF.	2, 4, 5
**Kidney biomarkers**		
Serum creatinine	Produced by skeletal muscle, its clearance its representative of renal function.	1, 2, 3, 4, 5
CysC	Cysteine proteinase inhibitor filtered through the glomerulus and then reabsorbed by tubular cells. It is an accurate surrogate marker of GFR.	1, 2, 3, 4, 5
Albuminuria	Marker of glomerular integrity/PCT function.	2, 4
TIMP/IGFBP7	Involved in G1 cell cycle arrest; it increases in tubular cell injury as an early marker.	1, 3, 5
NGAL	Small protein freely filtered through the glomerulus and reabsorbed in the proximal tubule. It increases in case of tubular damage 24 h before the rise of creatinine.	1, 3, 5
NAG	NAG is a lysosomal protein excreted into urine in case of tubular damage. NAG is increased in patients with AKI, CKD, or HF and may predict prognosis in these patients.	1, 3, 5
KIM1	KIM1 is expressed in proximal tubule cells after hypoxic injury and may identify the development of AKI or CKD in patients with HF. KIM1 is associated with HF, cardiovascular events, and deaths in patients with AKI and CKD.	1, 3, 5
IL-18	It is a component of NLEP3 inflammasome, and it is elevated in AKI. IL-18 levels are also increased in HF.	1, 3, 5
L-FABP	L-FABP is expressed in tubular epithelial cells and is excreted into urine with cytotoxic lipids. Urinary L-FABP has been associated with ischemic tubular injury and risk for acute kidney failure in type 1 CRS.	1, 3
α-1 Microglobulin	It is filtered by glomerulus and is completely reabsorbed by the renal tubule. It can be found in urine in case of tubule damage.	2, 4, 5

ACS, acute coronary syndrome; AHF, acute heart failure; AKI, acute kidney injury; CKD, chronic kidney disease; CRS, cardiorenal syndrome; cTn, cardiac troponin; CysC, cystatin C; GFR, glomerular filtration rate; HF, heart failure; H-FABP, heart-type fatty acid–binding protein; IGFBP7, insulin-like growth factor protein 7; IL, interleukin; KIM-1, kidney injury molecule-1; L-FABP, liver-type fatty acid–binding protein; MR-adrenomedullin, Mid-Regional- adrenomedullin; NAG, N-acetyl-κ-d-glucosaminidase; NGAL, neutrophil gelatinase-associated lipocalin; PCT, proximal convoluted tubule; PDGF, Platelet Derived Growth Factor; sFlt-1, soluble fms-like tyrosine kinase 1; sST2, soluble suppressor of tumorigenicity; Th2, T-helper type 2; TIMP, tissue inhibitor of metalloproteinase; VEGF, Vascular Endothelial Growth Factor.

**Table 3 ijms-24-05089-t003:** Therapeutic approaches for cardiorenal syndrome.

Drugs	Mechanism of Action	Side Effects/Contraindications	Clinical Use
Beta Blockers	Beta adrenergic Receptor Antagonism	Bradycardia, AVB; asthma/bronchospasm, hypotension, unstable HF.	Predominantly in chronic HF with reduced and mid-range ejection fraction without cardiogenic shock to improve morbidity and mortality. (CRS 2-4)
ACEi/ARB	Inhibition of ACE or AT1 receptor antagonism	Hyperkalemia, Renal failure, Hypotension, Idiopathic angio-oedema; Pregnancy, breastfeeding	Predominantly in chronic HF with reduced and mid-range ejection fraction without cardiogenic Shock to improve morbidity and mortality. (CRS 2-4)
ARNI	AT1/Neprilysin Inhibitor	Hyperkalemia, Renal Failure, Hypotension	Predominantly in chronic heart failure with reduced and mid-range ejection fraction without cardiogenic Shock to improve morbidity and mortality. (CRS 2-4)
MRA	Antagonism of mineralocorticoid receptor	Hyperkalemia, Renal Failure, Hypotension	Predominantly in chronic heart failure with reduced and mid-range ejection fraction without cardiogenic Shock to improve morbidity and mortality. (CRS 2-4)
SGLT2i	Antagonism of the cotransporter SLC5A2 in the PT1	Type 1 diabetes mellitus, Acute Metabolic Acidosis	Predominantly in chronic heart failure with reduced, ejection fraction without cardiogenic Shock to improve morbidity and mortality. (CRS 2-4)
Diuretics	NKCC, NCC, CA Antagonism	Hypotension, Hypokalemia, Hypo-/Hypercalcemia, Hyponatremia, Hypochloremia, Hypovolemia, Metabolic Alkalosis, Diuretic Resistance.	In Acute and Chronic heart failure with reduced, mid-range and preserved ejection fraction to improve symptoms and volume overload. (CRS 1-2-4) In acute and chronic kidney disease to maintain an effective diuresis (CRS 1-3-4)
Vaptans	Selective Antagonism of V2 Receptor	Pollakiuria, Nycturia, polydipsia, Hypernatremia, Signs of Liver Injury.	Advanced Heart Failure with hyponatremia (CRS 1-3)
Inotropic Drugs	Beta receptor agonism, Calcium sensitizers	Supraventricular and ventricular arrhythmias, increased myocardial oxygen consumption due to increased myocardial work, hypotension.	Acute heart failure and cardiogenic Shock (CRS 1)
UF/CRRT	Convection/Diffusion fluid and solute removal for the improvement of volume, osmolite and water balance	Hypotension, hypovolemia, reduced pre-load, thrombus formation, bleedings, vascular complications of the access site	Acute heart failure and cardiogenic shock in patients with volume overload in the absence of proper diuretic response (CRS 1-3)

ACE, Angiotensin Converting Enzyme; ARB, Angiotensin Receptor Blockers; ARNI, Angiotensin Receptor/Neprilysin Inhibitor; AVB, Atrioventricular Blocks; AT 1, Angiotensin Receptor 1; CA, Carbonic Anhydrase; CRRT, Continuous Renal Replacement Therapy; CRS, Cardio-renal Syndrome; HF, Heart Failure; MRA, Mineral corticoid Receptor Antagonists; NCC, Sodium Chloride Co-transporters; NKCC, Sodium Potassium Chloride Cotransporters; PT1, Proximal Tubule 1; SCLA5A2, Solute Carrier Family 5 Member 2; SGLT2, Sodium Glucose Co-Transporter 2; UF, Ultrafiltration.

## Data Availability

There are no new data associated with this article.

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
