# Peer review of "New Insight in Cardiorenal Syndrome: From Biomarkers to Therapy"

_ijms, 2023, doi:10.3390/ijms24065089_

Round 1
Reviewer 1 Report
Gallo et al. focus on three aspects of the current knowledge of cardiorenal syndrome: 1) the pathophysiology; 2) the utility of biomarkers; and 3) novel therapeutics for the management of CRS patients. This is a remarkable piece of work and I only have a few minor suggestions.
In Table 1, please describe the characteristics and mechanisms of each type of CRS. Also, it is clearer to provide clinical conditions of the renal syndrome and cardiac syndrome in separate columns.
Several biomarkers for Type 1 CRS are not discussed, such as Plasma proenkephalin A in opioid receptor-mediated negative inotropic effects. It is inversely related to GFR. Also with urinary angiotensinogen.
In the section for novel therapeutics, please also include device based therapy such as cardioverter defibrillator.
Duplication in references are identified. For example, Ref 56 and Ref 58.
Author Response
We thank the reviewer for her/his evaluation of our manuscript and her/his valuable comments. Appropriate changes have been made in the text as suggested by the reviewer and we now believe that the manuscript is improved.
Responses to reviewer’s comment are detailed below.
Q- In Table 1, please describe the characteristics and mechanisms of each type of CRS. Also, it is clearer to provide clinical conditions of the renal syndrome and cardiac syndrome in separate columns.
R- Thank you for this comment; we agree with the reviewer, and we have changed Table 1 according with reviewer's suggestions.
Please note that the query of this reviewer is similar to the one raised by reviewer # 4. We have changed the structure of Table 1 according to the suggestions of both reviewers. Thus, we have also removed the text in the section 2. Classification of Cardiorenal Syndrome which appeared repetitive of the table's contents.
Q-Several biomarkers for Type 1 CRS are not discussed, such as Plasma proenkephalin A in opioid receptor-mediated negative inotropic effects. It is inversely related to GFR. Also with urinary angiotensinogen.
R-We thank the reviewer for this comment. We have added further markers as correctly suggested. Please see the amended text in lines 432-442 in the revised manuscript.
Q- In the section for novel therapeutics, please also include device based therapy such as cardioverter defibrillator.
R- We expanded on device-based therapy, as correctly suggested by the reviewer. We now added a section with subheading "5.6 Non-pharmacological approaches" in lines 645-664 in the revised manuscript.
Q- Duplication in references are identified. For example, Ref 56 and Ref 58.
R- Thank you to rise this comment. We double checked the manuscript for reference duplication and we made changes accordingly.
Reviewer 2 Report
Is a good review.
Minor changes:
For Introduction author can divide it become etiology and epidemiology parts.
Please add subheader in the long area such as : pathophysiology of cardiorenal syndrome, biomarkers, therapeutic strategies.
Please add the prognosis information for cardiorenal syndrome
Author Response
We thank the reviewer for her/his evaluation of our manuscript and her/his valuable comments. Appropriate changes have been made in the text as suggested by the reviewer and we now believe that the manuscript is improved.
Responses to reviewer’s comment are detailed below.
Q-For Introduction author can divide it become etiology and epidemiology parts.
R-Thank you for this comment, we have rearranged the introduction as suggested by the reviewer and we also made changes by incorporating the comments of the Editorial Office.
Q- Please add subheader in the long area such as : pathophysiology of cardiorenal syndrome, biomarkers, therapeutic strategies.
R- We have modified the text as suggested by the reviewer. We modifyed the headings and we added subheadings. We have rearranged the revised manuscript as follow:
- Introduction
- Classification of Cardiorenal Syndrome
- Pathophysiology of cardiorenal syndrome
3.1 Hemodynamic factors
3.2 Endothelial dysfunction, oxidative stress, and inflammation
3.3 micro RNAs
- Biomarkers in cardiorenal syndrome
4.1 Cardiac biomarkers
4.2 Renal biomarkers
- Therapeutic strategies in CRS
5.1 Diuretic and ultrafiltration therapy
5.2 Inotropic agents and beta blockers
5.3 Renin angiotensin system inhibitors
5.4 SGLT2 inhibitors: an emerging therapeutic tool in CRS
5.5 Novel therapeutic strategies
5.6 Non-pharmacological approaches
- Conclusions
Q-Please add the prognosis information for cardiorenal syndrome.
R- We further expanded on the prognosis in the introduction as well as in the conclusions. The prognostic information are also present in the biomarkers' chapter. We also reinforced the concept of the utility of new drugs such as SGLT2 inhibitors in terms of efficacy on the prognosis in CRS patients (please see the section on therapies and conclusions).
Reviewer 3 Report
This review nicely summarizes the current knowledge on the pathophysiology of the cardiorenal syndrome (CRS). Especially biomarkers are listed and described in detail in terms of utility and usefulness for the different CRS classes. Furthermore therapeutic options are discussed. the article is interesting for abroad readership, clinicians and researchers. To my knowledge there is no comparable review review in the recent literature. the review is well andcomprensively written. however, there are a couple of minor concerns, which should be addressed to before the article could be accepted:
1) language, grammar, typing errors, definitions
a) line 31 replace "common ground" by "coomon basis"
b) line 129 GFR should be defined
c) line 144 "shown" instead of "showed"
d) line 175 "an" instead of " and"
e) line 212 " an important role" could replaced by " are pivotal" to avoid repetitions.
f) line 440 replace "aswell" by either "and" or " in addtion"
g) line444 "consist" instead of "consists"
h) line 500 a blank is missing between "with" and "high-dose"
2) more severe concerns
a) table 1 type I Do you mean acute HF or acute decompensated HF (ADHF) as written in the text line 68. These are two different disease conditions: acute HF might be caused by ischemia, arrhythmia, sudden increase in filling pressure etc. ADHF is characterized by a sudden worsening of chronic HF. PLease clarify and also add some charactersitics of AHF, chronic HF and ADHF.
b) line 148f the sentence is not qite clear Did you mean:" .......the increased IAP is strongly associated with renal dysfunction? Could you give a reference, please ?
c) line 164 ".. sugested by several evidences"Which? please, add references
d) paragraph on NO bioavailability lines175-188 increased ROS production is also happening CVDs. the oxidation of co factors of NOS reduce the bioavailability of NO and cGMP signalling in cardiomyocytes which promotes contractile dysfunction and remodelling. This should be somehow integrated
e) lines 273f Is alpha smooth muscle actin really expressed in diseased cardiomyocytes? To my knowledge not, but the expression of skeletal muscle actin is increased
f) pediatric cardiorenal syndrome is not mentioned, though causing a high mortality in these patients. my be you could add a few sentenes concerning this matter. do you know if there are any specities known concerning biomarkrs or therapeutic options?
Author Response
We thank the reviewer for her/his evaluation of our manuscript and her/his valuable comments. Appropriate changes have been made in the text as suggested by the reviewer and we now believe that the manuscript is improved.
Responses to reviewer’s comments are detailed below.
Q- 1) language, grammar, typing errors, definitions
- a) line 31 replace "common ground" by "coomon basis"
- b) line 129 GFR should be defined
- c) line 144 "shown" instead of "showed"
- d) line 175 "an" instead of " and"
- e) line 212 " an important role" could replaced by " are pivotal" to avoid repetitions.
- f) line 440 replace "aswell" by either "and" or " in addtion"
- g) line444 "consist" instead of "consists"
- h) line 500 a blank is missing between "with" and "high-dose"
R- We double checked for language, grammar, typos, and definitions thoughtout the manuscript. In particular, we amended the text according to the reviewer’s suggestions. Please note that in several parts the text has been modified in order to incorporate also the suggestions of the other reviewers and the Editorial Office's suggestions.
Q- 2) more severe concerns
Q- a) table 1 type I Do you mean acute HF or acute decompensated HF (ADHF) as written in the text line 68. These are two different disease conditions: acute HF might be caused by ischemia, arrhythmia, sudden increase in filling pressure etc. ADHF is characterized by a sudden worsening of chronic HF. PLease clarify and also add some charactersitics of AHF, chronic HF and ADHF.
R- We agree with the reviewer that acute HF is formally different by acute decompensated HF. Nevertheless, both conditions are characterized by acute reduction of cardiac function that may lead to renal impairment. Indeed, it has been reported that: “Acute heart failure (AHF) is a syndrome defined as the new onset (de novo heart failure (HF)) or worsening (acutely decompensated heart failure (ADHF)) of symptoms and signs of HF, mostly related to systemic congestion. In the presence of an underlying structural or functional cardiac dysfunction (whether chronic in ADHF or undiagnosed in de novo HF), one or more precipitating factors can induce AHF, although sometimes de novo HF can result directly from the onset of a new cardiac dysfunction, most frequently an acute coronary syndrome. Despite leading to similar clinical presentations, the underlying cardiac disease and precipitating factors may vary greatly and, therefore, the pathophysiology of AHF is highly heterogeneous. Left ventricular diastolic or systolic dysfunction results in increased preload and afterload, which in turn lead to pulmonary congestion. Fluid retention and redistribution result in systemic congestion, eventually causing organ dysfunction due to hypoperfusion”.(Please see: Arrigo m, Jessup m, Mullens w, Reza n, Shah AM, Sliwa K, Mebazaa A. Nat Rev Dis Primers 2020 Mar 5;6(1):16.
More recently, the modern classification of CRS has identified distinct groups in which acute heart failure is referred to a condition in which the acute loss of cardiac function (including cardiogenic shock) may lead to the acute decline in kidney function (type 1 CRS). Therefore, in table 1 we are referring to this condition in which the acute loss of cardiac function may contribute to the acute loss of renal function (type 1 -acute- CRS). On the other hand, in the setting of acute kidney insufficiency, acute heart failure may occur (type 3 CRS).
We have modified the manuscript accordingly to avoid confusion as correctly suggested.
Please note that the characteristics of all CRS types are now listed only in Table 1 which has been modified as suggested by the Reviewers # 1 & 4. Also the text of the section 2. Classification of Cardiorenal Syndrome has been removed by the manuscript to avoid repetitions of the table's content.
We also added the references 1,4, 9-15 in Table 1 of the revised manuscript.
Q- b) line 148f the sentence is not quite clear Did you mean:" .......the increased IAP is strongly associated with renal dysfunction? Could you give a reference, please ?
R- We have added the following reference in the manuscript in order support the concept that elevated IAP is associated with impaired renal function.
REF 26 in the revised manuscript: Wilfried Mullens, Zuheir Abrahams, Hadi N Skouri, Gary S Francis, David O Taylor, Randall C Starling, Emil Paganini, W H Wilson Tang Elevated intra-abdominal pressure in acute decompensated heart failure: a potential contributor to worsening renal function? J Am Coll Cardiol 2008 Jan 22;51(3):300-6)
Q- line 164 ".. sugested by several evidences"Which? please, add references
R- We added the following references (30-32 in the revised manuscript):
- Clementi, A.; VirziÌ€, G.M.; Battaglia, G.G.; Ronco, C. Neurohormonal, Endocrine, and Immune Dysregulation and Inflammation in Cardiorenal Syndrome. Cardiorenal Med. 2019, 9, 265–273
- Prastaro, M., Nardi, E., Paolillo, S., Santoro, C., Parlati, A. L. M., Gargiulo, P., Basile, C., Buonocore, D., Esposito, G., & Filardi, P. P. Cardiorenal syndrome: Pathophysiology as a key to the therapeutic approach in an under-diagnosed disease. Journal of clinical ultrasound : JCU, 2022, 50(8), 1110–1124
- Jentzer, J. C., & Chawla, L. S. A Clinical Approach to the Acute Cardiorenal Syndrome. Critical care clinics, 2015, 31(4), 685–703.
Q- d) paragraph on NO bioavailability lines175-188 increased ROS production is also happening CVDs. the oxidation of co factors of NOS reduce the bioavailability of NO and cGMP signalling in cardiomyocytes which promotes contractile dysfunction and remodelling. This should be somehow integrated
R - We agree with the reviewer and we expanded on the topic as follow (Please see also the revised text in lines 210-214):
Moreover, oxidative stress plays a central role for the signal transduction in cardiac cells in pathological conditions including heart failure. ROS induce inflammatory cytokines production, decrease NO–cyclic guanosine monophosphate signaling, impair endothelial function and induce MAP kinase which are all involved in cardiac hypertrophy and remodeling as well as myocardial dysfunction [48,49]."
Q- e) lines 273f Is alpha smooth muscle actin really expressed in diseased cardiomyocytes? To my knowledge not, but the expression of skeletal muscle actin is increased
R- Thank you for raising this doubt, in fact the expression of alpha-smooth muscle actin (a-SMA) protein is induced in cardiac myofibroblasts.
We have specified this in the revised manuscript.
Q- f) pediatric cardiorenal syndrome is not mentioned, though causing a high mortality in these patients. my be you could add a few sentenes concerning this matter. do you know if there are any specities known concerning biomarkrs or therapeutic options?
R- We agree with the reviewer that pediatric CRS is an important different entity that is attracting a growing interest, however this review article is focused only on CRS in adults (we further specified this focus in the abstract and in the introduction). Therefore, we did not discuss about pediatric CRS. There are other sources that specifically and extensively discussed this topic. Nevertheless, as suggested by the reviewer, we added few sentences on pediatric CRS in the revised manuscript (Please see lines 66-69 in the introduction)
Reviewer 4 Report
This is an interesting review manuscript with an important focus on cardiorenal syndrome. The authors are, however, suggested to pay attention to the following comments.
1) Page 3. Line 67-96: can you please include all your comments about the type 1 to 5 cardiorenal syndrome into the table 1? It would give more power to the table and it will not look so repetitive as it does now. Also include please the references.
2) Page 4 Figure 1: can you please remove the word “legend” as it is well-known that this is a legend to a figure. Please develop a bit more and explain in the legend the significance of the figure as well, not only some abbreviations. The same for Figure 2, page 17.
3) Can you please restructure the chapter 3 into subchapters such as ROS, VEGFS or miRNAs roles?
4) Can you please restructure the chapter 3 into subchapters such as “Cardiac “ and “Renal” biomarkers” as in the table 2? It is very hard to follow otherwise.
5) Can you please restructure chapter 5 by stating first which are the most important and used therapeutics and why and then explain the organization of this chapter – drugs potential and which novel drugs are foreseen using more recent studies.
Author Response
We thank the reviewer for her/his evaluation of our manuscript and her/his valuable comments. Appropriate changes have been made in the text as suggested by the reviewer and we now believe that the manuscript is improved.
Responses to reviewer’s comments are detailed below.
Q- Page 3. Line 67-96: can you please include all your comments about the type 1 to 5 cardiorenal syndrome into the table 1? It would give more power to the table and it will not look so repetitive as it does now. Also include please the references.
R- Thank you for this comment; we agree with the reviewer, and we have changed Table 1 according with the reviewer’s suggestions.
Please note that the comment of this reviewer is similar to the one raised by reviewer #1. Therefore, we have made changes according to the suggestiong of both reviewers. We have also removed the text in the section 2. Classification of Cardiorenal Syndrome which appeared repetitive of the table's content. We also added references [1,4,9-15] in table 1.
Q- Page 4 Figure 1: can you please remove the word “legend” as it is well-known that this is a legend to a figure. Please develop a bit more and explain in the legend the significance of the figure as well, not only some abbreviations. The same for Figure 2, page 17.
R- We have changed the figures according to the reviewer's suggestions. We have removed the word "Legend" which has been previously added according to the editorial rules.
Q- Can you please restructure the chapter 3 into subchapters such as ROS, VEGFS or miRNAs roles?
R- We agree with this suggestion. We restructured chapter 3 by adding sub-headings. Chapter 3 in the revised manuscript has now been restructured as follow:
- Pathophysiology of cardiorenal syndrome
3.1 Hemodynamic factors
3.2 Endothelial dysfunction, oxidative stress, and inflammation
3.3 micro RNAs
Q- Can you please restructure the chapter 3 into subchapters such as “Cardiac “ and “Renal” biomarkers” as in the table 2? It is very hard to follow otherwise.
We agree with this suggestion again. We restructured chapter 4 by adding sub-headings. Chapter 4 in the revised manuscript has now been restructured as follow:
- Biomarkers in cardiorenal syndrome
4.1 Cardiac biomarkers
4.2 Renal biomarkers
Q- Can you please restructure chapter 5 by stating first which are the most important and used therapeutics and why and then explain the organization of this chapter – drugs potential and which novel drugs are foreseen using more recent studies.
R- We have modified the chapter as suggested. Chapter 5 in the revised manuscript has now been restructured with the following heading and sub-headings:
- Therapeutic strategies in CRS
5.1 Diuretic and ultrafiltration therapy
5.2 Inotropic agents and beta blockers
5.3 Renin angiotensin system inhibitors
5.4 SGLT2 inhibitors: an emerging therapeutic tool in CRS
5.5 Novel therapeutic strategies
5.6 Non-pharmacological approaches
Please see also the amended text of the revised manuscript